# An N-terminal–truncated isoform of FAM134B (FAM134B-2) regulates starvation-induced hepatic selective ER-phagy

Shohei Kohno, Yuji Shiozaki, Audrey L Keenan, Shinobu Miyazaki-Anzai, Makoto Miyazaki ⓘ

**Autophagy is a conserved system that adapts to nutrient starvation, after which proteins and organelles are degraded to recycle amino acids in response to starvation. Recently, the ER was added to the list of targets of autophagic degradation. Autophagic degradation pathways of bulk ER and the specific proteins sorted through the ER are considered key mechanisms in maintaining ER homeostasis. Four ER-resident proteins (FAM134B, CCPG1, SEC62, and RTN3) have been identified as ER-resident cargo receptors, which contain LC3-interacting regions. In this study, we identified an N-terminal–truncated isoform of FAM134B (FAM134B-2) that contributes to starvation-induced ER-related autophagy. Hepatic FAM134B-2 but not full-length FAM134B (FAM134B-1) is expressed in a fed state. Starvation drastically induces FAM134B-2 but no other ER-resident cargo receptors through transcriptional activation by C/EBPβ. C/EBPβ overexpression increases FAM134B-2 recruitment into autophagosomes and lysosomal degradation. FAM134B-2 regulates lysosomal degradation of ER-retained secretory proteins such as ApoCIII. This study demonstrates that the C/EBPβ-FAM134B-2 axis regulates starvation-induced selective ER-phagy.**

## Introduction

The ER is the most abundant membrane structure in the cell. The ER is the central organelle that regulates protein synthesis and modification, lipid metabolism, and calcium homeostasis (Borgese et al, 2006; Braakman & Hebert, 2013; Krebs et al, 2015). Differences in physical and functional characteristics distinguish the two types of ER, known as rough ER (RER) and smooth ER (SER). RER attaches to ribosomes to synthesize proteins, whereas SER synthesizes lipids (Borgese et al, 2006). Two major pathways in the ER, the unfolded protein response and ER-associated protein degradation (ERAD), are known to control ER homeostasis when ER perturbations occur. Unfolded protein response activation increases the folding capacity of the ER, whereas the ERAD system recognizes terminally misfolded proteins and degrades

them by the ubiquitin-proteasome system (Hwang & Qi, 2018; Kroeger et al, 2018). These two pathways maintain the flow of synthesis, folding and clearance of ER-resident proteins. In addition to the two ER homeostasis pathways, recent studies revealed the ER as a new target of autophagic degradation (Mochida et al, 2015). ER-resident membrane proteins family with sequence similarity 134 member B (FAM134B), preprotein translocation factor (SEC62), cell cycle progression protein 1 (CCPG1), and reticulon 3 (RTN3) were identified as ER-phagy receptors (Khaminets et al, 2015; Fumagalli et al, 2016; Grumati et al, 2017; Fregno & Molinari, 2018; Smith et al, 2018). These proteins contain LC3-interacting regions in a cytosolic domain and mediate autophagic cargo followed by lysosomal degradation.

The liver is a central organ that regulates nutrient and drug metabolisms in response to various nutritional stresses. Nutrient starvation is a major trigger regulating several adaptive changes in the liver, including macroautophagy (Komatsu et al, 2005; Ezaki et al, 2014; Fullgrabe et al, 2016). Starvation-induced hepatic autophagy is regulated by transcription factors and nutrient-sensing kinases at the transcriptional and post-translational levels, respectively (Goldstein & Hager, 2015; Ueno & Komatsu, 2017). At the transcriptional level, a number of transcription factors contribute to starvation-induced autophagy gene expression, such as *CREB*, *PPARs*, *FXR*, *FOXOs*, and *C/EBPs* (Goldstein & Hager, 2015). Starvation is also known to induce autophagy-mediated ER degradation (Komatsu et al, 2005). However, which ER-resident cargo receptor contributes to starvation-induced hepatic ER-phagy and which transcription factor regulates starvation-induced ER-related selective autophagy have not been elucidated.

In this study, we identified a novel N-terminal–truncated isoform of FAM134B (FAM134B-2) as a starvation-induced ER-resident cargo receptor. Starvation selectively induced FAM134B-2 gene expression through the induction of C/EBPβ and also increased the recruitment of FAM134B-2 into autophagosomes and lysosomal degradation of FAM134B-2 in vivo and in vitro. FAM134B-2 was localized in both the SER and RER. In addition, FAM134B-2 regulates the selective ER-phagy of secretary proteins such as apolipoprotein C-III (ApoCIII) but not bulk ER turnover. These data suggest that FAM134B-2 is a major contributor to starvation-induced selective ER-phagy.

Division of Renal Diseases and Hypertension, Department of Medicine, University of Colorado Denver, Aurora, CO, USA

Correspondence: makoto.miyazaki@ucdenver.edu

# Results

## Selective induction of a truncated isoform of FAM134B (FAM134B-2) by starvation

We performed quantitative reverse transcription PCR (qRT-PCR) to determine which hepatic ER-resident cargo receptor is transcriptionally regulated upon starvation. 16 h of fasting selectively increased mRNA levels of *FAM134B*, which were normalized by 6 h of re-feeding (Fig 1A). Hepatic mRNA expression of the other ER-phagy receptors (SEC62, CCPG1, and RTN3) was not altered by fasting and re-feeding (Fig 1A). We next determined whether starvation increases levels of FAM134B protein. To identify FAM134B protein-specific signal in the liver, we used two references: 1) protein lysates from the mouse brain, which highly expresses FAM134B and 2) recombinant mouse FAM134B protein generated in HEK293T cells. Immunoblot analysis showed that the mouse liver did not express full-length FAM134B (FAM134B-1) in either the fed or fasted states,

as described recently by Schultz et al (2018). Interestingly, however, the smaller size of FAM134B (FAM134B-2) was only detected in the livers of fasted mice (Fig 1B). Although the expected sizes of FAM134B-1 and FAM134B-2 are ~50 and ~39 KD, respectively, on the SDS–PAGE, FAM134B-1, and FAM134B-2 migrated to ~65 and ~50 KD, respectively. This is probably due to post-translational mechanisms. The antibody recognizes the C-terminal region of the FAM134B protein, suggesting that the truncation occurs at the N-terminal of the protein. Because FAM134B-1 was not detected in the liver, the truncation was probably not due to the post-translational proteolytic cleavages. We, therefore, hypothesized that another isoform of *FAM134B* mRNA occurs through alternative transcription. To identify the alternative transcript of *FAM134B*, we performed 5'-rapid amplification of cDNA ends (RACE) using cDNA from fasted mouse liver; brain cDNA was used as a reference. The 5'RACE analysis showed that liver and brain cDNA produced different sizes of the N-terminal *FAM134B* DNA fragments (Fig 1C). The cDNA sequences of *FAM134B* amplified by 5'RACE are shown in

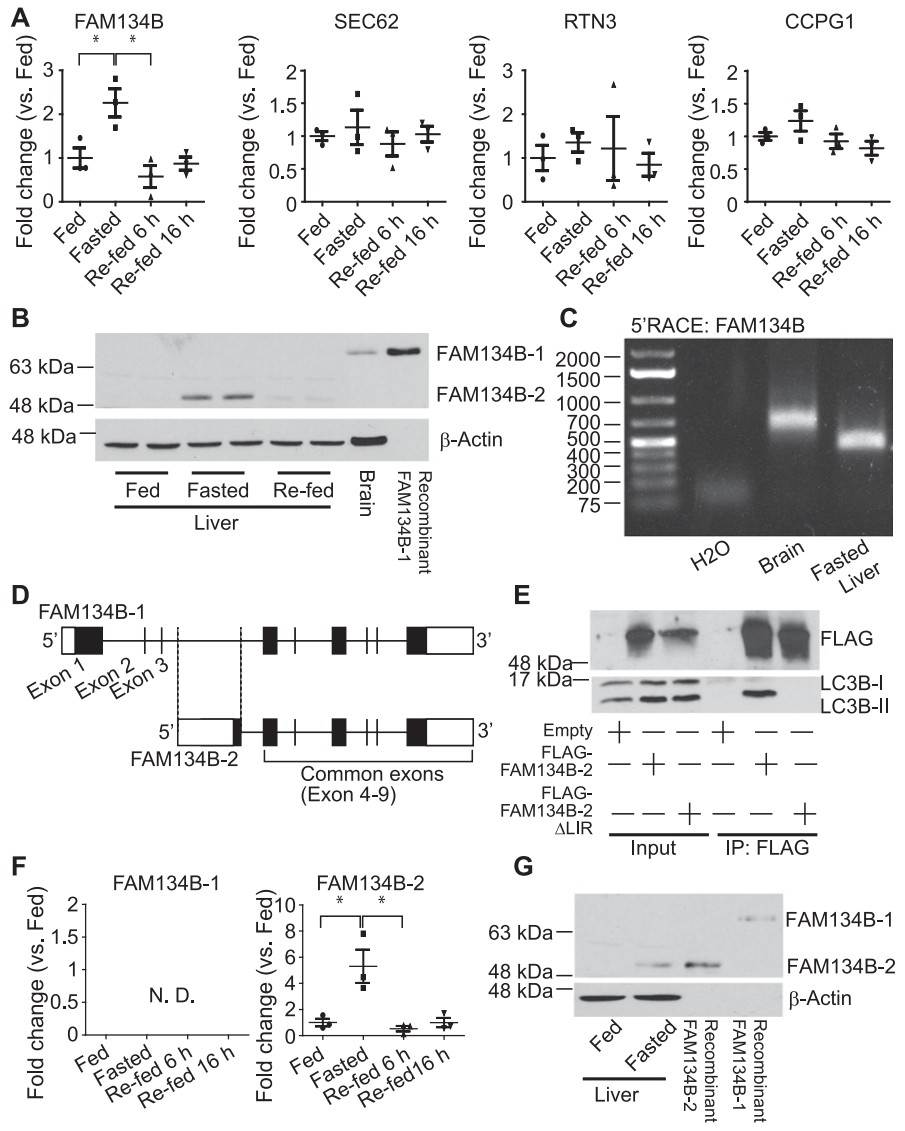

**Figure 1. Starvation triggers the selective induction of FAM134B-2.**
**(A)** qRT-PCR analysis of FAM134B, SEC62, RTN3, and CCPG1 in mouse livers from fed, fasted, and re-fed conditions. **(B)** Immunoblot analysis of FAM134B in mouse livers (fed, fasted, and re-fed), mouse brain, and recombinant FLAG-FAM134B-1 generated in HEK293T cells. **(C)** Gel electrophoresis of 5' RACE products using brain cDNA from fed mice or liver cDNA from fasted mice. **(D)** Schematic models of mouse FAM134B-1 and FAM134B-2 genes. **(E)** Co-immunoprecipitation of endogenous LC3B, FLAG-FAM134B-2, and FLAG-FAM134B-2-ΔLIR. FLAG-FAM134B-2 was transiently overexpressed in HEK293T cells. 24 h after transfection, the cells were starved for 6 h in the presence of EBSS and bafilomycin A1 (1 μM). **(F)** qRT-PCR analysis of FAM134B-1 and FAM134B-2 in mouse livers under fed, fasted, and re-fed conditions. **(G)** Immunoblot analysis of FAM134B in mouse livers (fed and fasted), recombinant FLAG-FAM134B-1 and 2 generated in HEK293T cells. One-way ANOVA with a Student–Newman post hoc test was used for statistical analysis. *$P < 0.05$.

Fig S1. Fig 1D shows the alternative forms of exon 1 in *FAM134B*. The novel truncated isoform of the *FAM134B* gene (*FAM134B-2*) consists of 6 exons which encode 356 amino acids, whereas *FAM134B-1* consists of 9 exons which encode 480 amino acids (Fig 1D). The exon 1 of *FAM134B-2* splices into the exon 4 of *FAM134B-1* (Fig 1D). An analysis using the network-based algorithm Protter (http://wlab.ethz.ch/protter/start/) predicted that FAM134B-2 is a type I membrane protein consisting of one transmembrane domain (amino acids 55–88), whereas FAM134B-1 has four transmembrane domains. More importantly, FAM134B-2 preserves the cytosolic LC3-interacting region but partially lacks the reticulon homology domain, both of which are essential for this function (Fig S2B) (Khaminets et al, 2015). The structures of the human FAM134B-2 gene and protein are very similar to those of mouse FAM134B-2 (Fig S2A and B). We, therefore, performed co-immunoprecipitation using FLAG-tagged FAM134B-2 expressing HEK293T cells to determine the interaction between FAM134B-2 and LC3B. The immunoprecipitant with FLAG antibody contains LC3B-II, whereas the deletion of the LIR of FAM134B-2 abolishes the interaction with LC3B-II (Fig 1E). The data suggest that FAM134B-2 interacts with LC3B-II and the reticulon homology domain may not be essential for this function. To separately quantify levels of *FAM134B* isoforms, we designed specific primers for each isoform as shown in Fig S2C and Table S1. The qRT-PCR primer pairs specifically amplified the target isoforms (Fig S2C). *FAM134B-1* mRNA expression was detected in the brain but not in the fasted liver, whereas *FAM134B-2* mRNA expression was detected in the liver but not in the brain (Fig S2C). Hepatic *FAM134B-1* mRNA was undetectable in both fed and fasted states (Fig 1F). Starvation selectively induced mRNA levels of hepatic *FAM134B-2*, consistent with the immunoblot analysis (Fig 1F). Re-feeding normalized starvation-induced *FAM134-2* expression (Fig 1F). The immunoblot analysis confirmed that the size of the smaller FAM134B signal induced by starvation exactly matched that of recombinant FAM134B-2 protein but not FAM134B-1 (Fig 1G). We next analyzed whether starvation induces FAM134B-2 in other organs. As shown in Fig S2D, starvation induced FAM134B-2 expression in peripheral organs such as the kidney, spleen, and white adipose tissue but not the brain, where FAM134B-1 is highly expressed.

## C/EBPβ mediates starvation-induced FAM134B-2 expression

Because levels of *FAM134B-2* mRNA were increased under starvation, we thought that starvation regulates *FAM134B-2* expression during transcription. To elucidate which starvation-regulating transcription factor induces *FAM134B-2* transcription, 1.5-kb *FAM134B-2* promoter was cloned to a luciferase reporter vector, pGL3, which was transfected onto HEK293T cells with 19 known transcription factors involved in fasting and starvation (Goldstein & Hager, 2015). As shown in Fig 2A, C/EBPα and C/EBPβ but not the other transcription factors significantly increased the promoter-driven luciferase activity by 16- and 9-fold, respectively, compared with empty vector. We next confirmed whether starvation induces levels of hepatic C/EBPα and C/EBPβ. qRT-PCR and immunoblot analysis showed that C/EBPβ (both liver-enriched transcriptional activator protein [LAP] and liver-enriched transcriptional inhibitory protein [LIP]) but not C/EBPα expression was induced in the livers of fasted mice (Fig 2B–F). Starvation-induced C/EBPβ expression was normalized by re-feeding

(Fig 2C, D, and F). Starvation-induced levels of C/EBPβ in the kidney, spleen, and white adipose tissue, whereas the brain expressed C/EBPβ at very low levels (Fig S3A). To find which region of the *FAM134B-2* promoter is responsible for C/EBPβ–mediated FAM134B-2 induction, a promoter deletion assay was performed. C/EBPβ overexpression increased 1.5, 0.5, and 0.2 kb *FAM134B-2* promoter-derived luciferase activity by sixfold, sixfold, and fourfold, respectively, whereas the 0.1-kb promoter was not activated (Fig 2G). These results suggest that the C/EBPβ–responsive element should be located between positions −500 to −100 of the *FAM134B-2* promoter. A Web-based computational analysis (www.gene-regulation.com) of the *FAM134B-2* promoter revealed three putative C/EBPβ–binding sites (Fig 2H). To demonstrate which site mediates the effect of C/EBPβ on the *FAM134B-2* promoter, mutations in all three putative sites were generated in the pGL3 containing 1.5-kb *FAM134B-2* promoter and subjected to the same transfection assay. Mutations in the −154 to −141 positions of the *FAM134B-2* promoter but no other mutations significantly reduced the response to C/EBPβ overexpression (Fig 2H). Together, these results indicate that a C/EBPβ responsive element is located in the *FAM134B-2* promoter between positions −154 to −141. To confirm that C/EBPβ physiologically binds to the C/EBPβ responsive element of the *FAM134B-2* promoter, we performed chromatin-immunoprecipitation (ChIP) assays in the livers from fed and fasted animals. The ChIP-RT-qPCR assay confirmed that starvation significantly increased the recruitment of C/EBPβ but not C/EBPα onto the *FAM134B-2* promoter in vivo compared with the fed state (Fig 2I and J). To examine whether C/EBPβ is fully responsible for the induction of hepatic FAM134B-2 expression, we generated liver-specific human Myc-C/EBPβ (also called LAP) transgenic mice (Fig 3A). To eliminate undesirable effects by random insertion of a transgene into the genome, floxed human Myc-C/EBPβ was inserted into a known permissive locus, Rosa26, by a recombinase-mediated cassette exchange procedure that we previously developed and have used for other transgenic lines. *Rosa-hC/EBPβ* mice were crossed with *Albumin (Alb)-Cre* mice. *Alb-Cre; Rosa-hC/EBPβ* mice expressed Myc-hC/EBPβ only in the liver (Fig S3B), which increased the levels of hepatic total C/EBPβ protein under the fed state compared with *Alb-Cre* (control) littermates (Fig 3C). The overexpression of hepatic C/EBPβ induced mRNA and protein expression of *FAM134B-2* (Fig 3B and C) but no other ER-phagy receptors (Fig S3C–F). C/EBPβ overexpression also increased levels of hepatic ATG5 and LC3B-II protein, suggesting that overall autophagy is induced by C/EBPβ overexpression (Figs 3C and S3G, and H). Injections of leupeptin (an inhibitor of lysosomal cysteine, serine, and threonine peptidases) increased FAM134B-2 protein levels in L-C/EBPβ mice (Fig 3D). To further confirm the C/EBPβ–mediated induction of FAM134B-2, we generated liver-specific C/EBPβ KO (Alb-Cre(+); C/EBPβ^flox/flox L-C/EBPβ KO) mice by crossing *Alb-Cre* and *C/EBPβ*–floxed mice. We used *Alb-Cre(−); C/EBPβ^flox/flox* mice as a control. Liver-specific *C/EBPβ* deficiency significantly blocked starvation-induced FAM134B-2 expression (Fig 3E and F).

## FAM134B-2 is expressed in the SER and RER

We next investigated whether FAM134B-2 protein is localized in a specific ER compartment using iodixanol density gradient ultracentrifugation (Opti-Prep) with hepatic microsomes of fasted mice. Sarco/endoplasmic reticulum $Ca^{2+}$-ATPase-2 (SERCA2) and

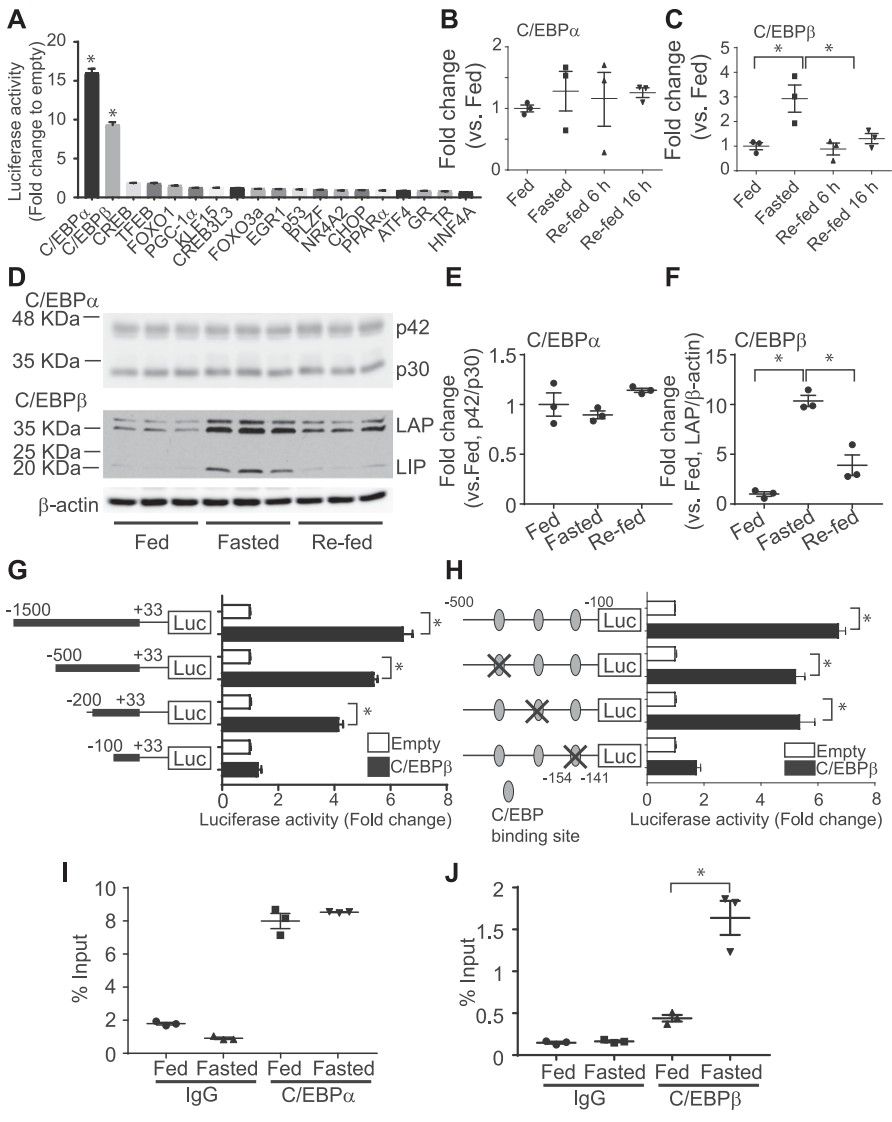

**Figure 2. C/EBPβ mediates FAM134B-2 transcription.**
**(A)** Promoter analysis of the mouse FAM134B-2 gene using a luciferase (Luc) reporter gene assay. HEK293T cells were co-transfected with firefly luciferase reporter plasmid containing the 1.5-kb promoter region of the mouse FAM134B-2 gene, pCMV-LacZ, and empty or indicated transcriptional factor expression plasmids. Results are expressed as the relative Luc/-galactosidase units of induction (n-fold) over the empty plasmid. **(B–D)** qRT-PCR analysis and (D) immunoblot analysis of C/EBPα and C/EBPβ in mouse livers from fed, fasted, and re-fed conditions. **(E, F)** Immunoblot results of (E) C/EBPα and (F) C/EBPβ were quantified using a densitometric analysis. **(G, H)** Deletion (G) and mutational (H) analysis of the mouse FAM134B-2 gene using a luciferase reporter gene assay. The schematic illustrations represent the serially deleted FAM134B-2/Luc reporter constructs (G) or mutated FAM134B-2/Luc reporter constructs (H). Results are expressed as the relative Luc/β-galactosidase units of induction (n-fold) over the empty plasmid. **(I, J)** ChIP assay for in vivo binding status of C/EBPα (I) or C/EBPβ (J) on the FAM134B-2 promoter. Mouse liver chromatin extracts from fed or fasted conditions were immunoprecipitated with anti-C/EBP antibodies or normal IgG. Purified DNA was determined with qRT-PCR. Data show percentage to input DNA. One-way ANOVA with a Student–Newman post hoc test was used for statistical analysis. *P < 0.05.

stearoyl-CoA desaturase (SCD1) were used as specific markers for SER, whereas ribosome-associated membrane protein-4 (RAMP4) and SEC61 were used as markers for RER. Opti-Prep was able to clearly separate SER and RER (Fig 4A). Interestingly, FAM134B-2 protein is more abundant in SER fractions than RER fractions, whereas FAM134C protein is equally detected in both SER and RER fractions. Another ER-resident cargo receptor, SEC62, is only found in RER fractions (Fig 4A). FAM134A, CCPG1, and RNT3 proteins were not detected in hepatic SER and RER, probably because of very low expression. Immunofluorescence confocal microscopic analysis confirmed that endogenous and exogenous FAM134B-2 is colocalized with both SERCA2 and SEC62 in HeLa cells (Fig S4A–C).

## Starvation increases lysosomal FAM134B-2 degradation in vivo and in vitro, and in mice and in humans

To examine whether starvation increases lysosomal ER protein degradation in vivo, we treated mice with a lysosome inhibitor, leupeptin (i.p., 15 mg/kg body weight) under fed and fasted states. Leupeptin is most commonly used as an inhibitor for hepatic autophagy (Haspel et al, 2011; Ezaki et al, 2014; Zhang et al, 2015; Moulis & Vindis, 2017). 4-h leupeptin treatment increased levels of FAM134B-2 protein under the fed state, but increased levels more significantly under the fasted state (Fig 4B and C). As expected, levels of LC3B-II protein were increased by leupeptin treatment, especially under the fasted state (Fig 4B and D). We next examined which ER protein affects starvation-induced ER-phagy. Interestingly, starvation did not change levels of SERCA2, RTN4, and CLIMP63, which are known as FAM134B-1 substrates (Fig 4B and E).

We next examined whether starvation-induced FAM134B and ER-related autophagy can be replicated in primary hepatocytes. However, primary mouse hepatocytes are not a suitable model to study FAM134B-2–mediated ER-related autophagy based on the following observations as shown in Fig S5: 1) Isolation of primary hepatocytes alone induced the expression of *FAM134B-2* by more than 50-fold compared with the intact liver (Fig S5A and B). 2)

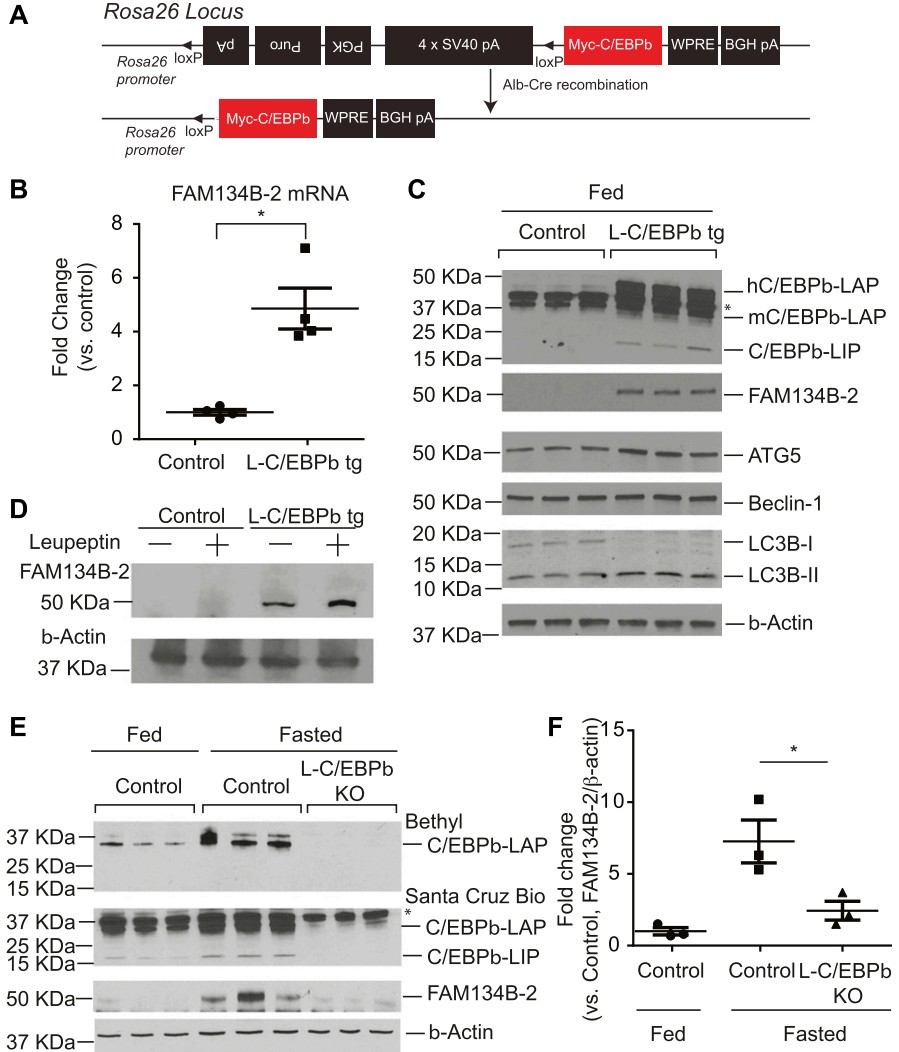

**Figure 3. C/EBPβ regulates FAM134B-2 and autophagy regulators in the mouse liver.**
**(A)** Scheme of construct design for targeting the C/EBPβ transgene. **(B)** qRT-PCR analysis of FAM134B-2 in livers from WT and L-C/EBPβ tg mice under a fed state. **(C)** Immunoblot analysis of FAM134B-2, ATG5, Beclin-1, and LC3B in livers from WT and L-C/EBPβ tg mice under a fed state. *Non-specific signal. **(D)** Immunoblot analysis of FAM134B-2 in livers from WT and L-C/EBPβ tg mice treated with leupeptin (i.p., 15 mg/kg body weight) for 4 h. **(E)** Immunoblot analysis of FAM134B-2 in livers of L-C/EBPβ KO mice under starvation. Mice were fasted for 16 h. **(F)** Quantitative densitometry of the immunoblot analysis. *Nonspecific signal. Student's unpaired t test was used for statistical analysis. *P < 0.05. Source data are available for this figure.

Treatment with a starvation medium (Earle's Balanced Salt Solution [EBSS]) did not affect levels of FAM134B-2. 3) Treatment of primary hepatocytes with the autophagy inhibitor bafilomycin A1 did not affect levels of FAM134B-2 (Fig S5C). As a control, LC3B-II protein was accumulated in primary hepatocytes treated with bafilomycin A1 (Fig S5C). The results suggest that the isolation of primary hepatocytes caused dysfunction of FAM134B-mediated ER-related autophagy.

We strived to find an in vitro system (i.e., cell line and medium) to study starvation-induced FAM134B-2 and ER-related autophagy and examine whether starvation induces FAM134B-2 in humans. Because starvation induced FAM134B-2 expression in not only the liver but also most peripheral tissues in vivo, we thought that some cell lines might respond to starvation and induce FAM134B-2 expression. Four human cell lines (HeLa, HEK293T, HepG2, and U2OS) and one mouse cell line (NIH3T3) were treated with a starvation media. In a complete media (DMEM containing 10% FBS), *FAM134B-2* mRNA was detected in HeLa cells but not HEK293T, U2OS, HepG2, or NIH3T3 cells (Fig S6A). Similar to in vivo starvation, HeLa cells starved with EBSS increased FAM134B-2 mRNA levels by fivefold compared with

complete media (DMEM containing 10% FBS), whereas levels of *FAM134B-1* mRNA were reduced in starvation media (Fig 5A and B). EBSS starvation also increased *FAM134B-2* mRNA levels in HEK293T cells but not U2OS or HepG2 cells (Fig S6A). Immunoblot analysis confirmed that in HeLa cells, EBSS treatment increased levels of FAM134B-2 protein, which was further increased by treatment with bafilomycin A1 (Fig 5C). Starvation also increased FAM134B-2 protein levels in HEK293T cells but not U2OS cells (Fig S6B). Starvation increased levels of C/EBPβ protein in HeLa cells but not HepG2 cells (Fig 5C). We also tested whether C/EBPβ induces FAM134B-2 expression in HeLa cells. C/EBPβ overexpression significantly induced FAM134B-2 expression in HeLa cells, consistent with the in vivo results (Fig S6C). We next investigated whether C/EBPβ induces the recruitment of FAM134B-2 into autophagosomes using immunofluorescence confocal microscopy analysis. HeLa cells were immunolabeled with antibodies targeting FAM134B and LC3. C/EBPβ overexpressing cells had remarkably wider colocalized areas of FAM134B and LC3 under bafilomycin A1 treatment (Fig 5D). We quantified the colocalization using Pearson correlation

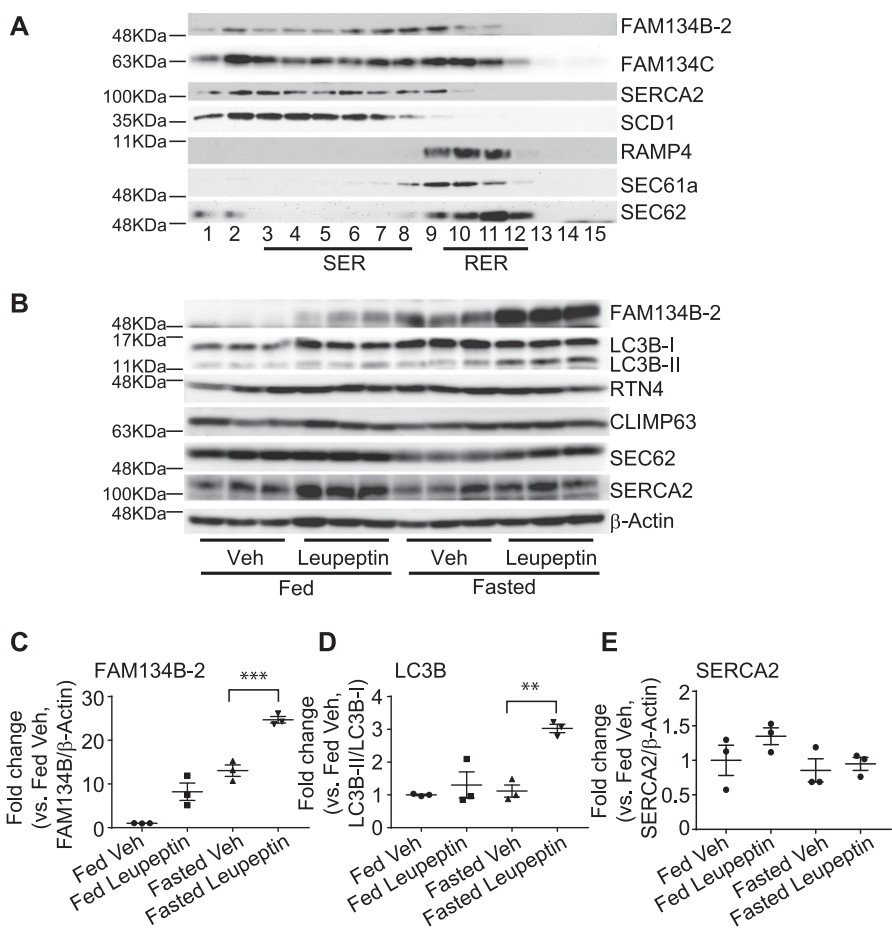

**Figure 4.   Hepatic FAM134B-2 is located in the SER and RER and degraded in the lysosome.**
**(A)** Immunoblot analysis of FAM134B-2, FAM134C, SERCA2, SCD1, RAMP4, SEC61a, and SEC62 in liver ER subfractions. Microsomes were isolated from fasted mice and separated to ER subfractions using iodixanol density gradient ultracentrifugation (Opti-Prep). **(B)** Immunoblot analysis of FAM134B-2, LC3B, RTN4, CLIMP63, SEC62, and SERCA2 in the mouse liver. Leupeptin (15 mg/kg body weight) or PBS was intraperitoneally injected into fed or fasted mice. After 4 h, the mice were sacrificed. **(C–E)** Densitometric analysis of FAM134B-2 (C), LC3B (D), and SERCA2 (E) in livers with leupeptin treatment. Student's unpaired *t* test was used for statistical analysis. *P < 0.05.

coefficient (PCC) analysis, which is widely accepted as a statistical measure of colocalization (McDonald & Dunn, 2013). PCC confirmed that C/EBPβ overexpression significantly induced FAM134B-2 and LC3 colocalization (Fig 5E). C/EBPβ overexpression also significantly increased LC3B expression (Fig 5F). Furthermore, this colocalization was observed in LAMP1 positive lysosomes (23.84% ± 3.03, N = 8, Fig 5G). These results suggest that starvation induces C/EBPβ-FAM134B-2–mediated ER-related autophagy in humans similar to mice.

### FAM134B-2 regulates selective degradation of ER-retained secretary proteins but not bulk ER

Because FAM134B-1 has been identified as an ER-phagy receptor that controls the size of the ER (Khaminets et al, 2015), we next examined whether FAM134B-2 regulates ER size in HeLa cells. Unexpectedly, *FAM134B* deficiency did not change ER size in HeLa cells (Fig S7A–C). FAM134B-2 reconstitution to *FAM134B* knockdown cells also did not change ER size. In addition, *FAM134B* knockdown did not change levels of ER structural proteins and chaperones in HeLa cells (Fig S7D). Furthermore, FAM134B-2 did not change the size of SEC61B-positive sheet-like cisternal ER and RTN4-positive tubular ER (Fig S7E–H). These data suggest that FAM134B-2 does not participate in bulk ER turnover. Recent studies suggest that

FAM134B-1 controls a selective ER-phagy in addition to ER-phagy (Forrester et al, 2018; Schultz et al, 2018). We strived to identify in vivo substrates for FAM134B-2 using *FAM134B* KO mice (Fig S8A). Since the liver specifically expresses FAM134B-2, ER fractions were isolated from the livers of fasted *FAM134B* KO and wild-type mice and subjected to a shotgun nontargeted proteomic analysis. The proteomic analysis detected 1,415 proteins and identified 40 proteins that were significantly increased in the microsomes from fasted *FAM134B* KO mice (Fig 6A) compared with the microsomes from fasted wild-type mice. In this study, we focused on ApoCIII because 1) it was consistently increased second-most in *FAM134B* KO liver microsomes, 2) it is a secretary protein modified in the ER (Khetarpal et al, 2017), 3) it is very abundant in the liver (Reue et al, 1988), and 4) a specific antibody is commercially available. Immunoblot analysis confirmed that levels of ApoCIII were significantly higher in the microsomal protein fractions from *FAM134B* KO mice (Figs 6B and S8B). mRNA levels of *ApoCIII* were similar in wild-type and *FAM134B* KO mice under fasting conditions (Fig S8C). Levels of ER-resident proteins such as Climp63, RTN4, BiP, and calnexin were not changed in the total protein and the microsomal protein fractions of livers from fasted *FAM134B* KO mice (Figs 6B and S8D). Starvation reduced protein levels of ApoCIII, whereas liver *C/EBPβ* deficiency increased levels of ApoCIII under fasting conditions (Figs 6C and S8E). In contrast, C/EBPβ overexpression in the liver resulted

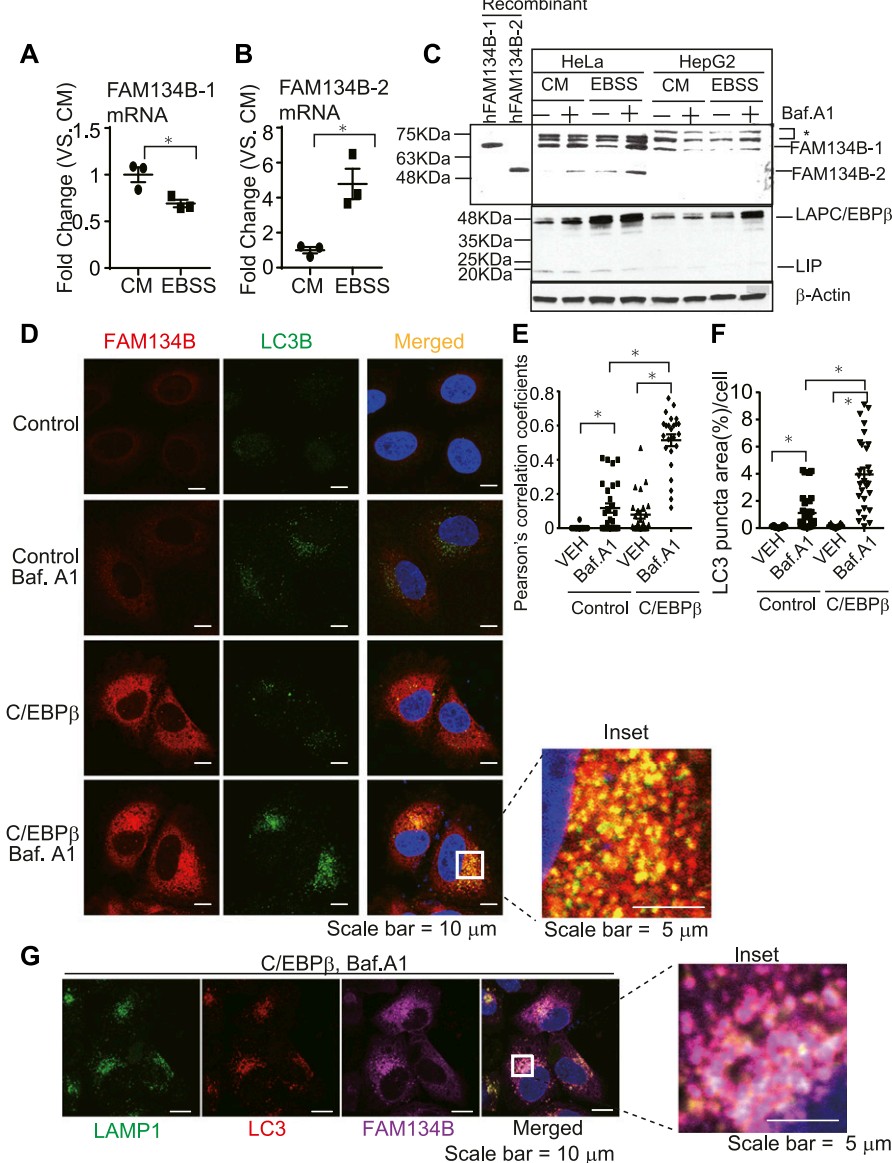

**Figure 5.    C/EBPβ induces FAM134B-2 expression, the recruitment of FAM134B-2 into autophagosomes, and lysosomal FAM134B-2 degradation.**
**(A, B)** qRT-PCR analysis of FAM134B-1 (A) and FAM134B-2 (B) in HeLa cells. Cells were cultured with complete media (CM) or EBSS for 6 h. **(C)** Immunoblot analysis of FAM134B-2 in HeLa and HepG2 cells. Cells were cultured with complete media (CM) or EBSS for 6 h in the albescence/presence of bafilomycin A1 (1 μM) for 6 h. *Nonspecific signal. **(D)** Representative images of immunofluorescence using HeLa cells. HeLa cells overexpressed C/EBPβ and were cultured with bafilomycin A1 or vehicle for 6 h. Green = LC3, red = FAM134B. **(E)** PCCs between FAM134B and LC3 were quantified. **(F)** Area of LC3 puncta was quantified. **(G)** Representative images of immunofluorescence using HeLa cells. HeLa cells overexpressed C/EBPβ and were cultured with bafilomycin A1 for 6 h. Green = LAMP1, red = LC3, magenta = FAM134B. One-way ANOVA with a Student–Newman post hoc test was used for statistical analysis. *P < 0.05.

in decreased levels of ApoCIII under fed conditions (Figs 6D and S8F). Immunofluorescence confocal microscopic analysis confirmed that *FAM134B* deficiency led to ApoCIII accumulation in the ER of HeLa cells (yellow), whereas FAM134B-2 reconstitution normalized ER-retained ApoCIII levels (Fig 7A–C). Co-immunoprecipitation revealed that FAM134B-2 interacted with ApoCIII under EBSS starvation conditions (Fig 7D). To examine whether lysosomal FAM134B-2 and ApoCIII degradation are ATG7-dependent, *ATG7* KO HeLa cells were generated (Fig S8G). Levels of FAM134B-2 and ApoCIII were significantly increased in *ATG7* KO cells (Fig S8G and H). *ATG7* deficiency also increased levels of ER-retained ApoCIII and blocked the delivery of ApoCIII to lysosomes (Fig S8I and J). Immunofluorescence confocal microscopy revealed that FAM134B-2 (green) colocalized with FAM134B-2-ApoCIII complexes (yellow) in the lysosome (magenta: ring-like structure) under starved conditions (Fig 7E and F). *FAM134B* deficiency failed to

deliver ApoCIII (red) to the lysosome (green: ring-like structures) and FAM134B-2 reconstitution normalized ApoCIII delivery to the lysosome (Fig 7G and H).

## Discussion

Starvation induces hepatic ER-phagy (Komatsu et al, 2005). Four ER-resident cargo receptors (FAM134B, CCPG1, RTN3, and SEC62) have been identified so far (Khaminets et al, 2015; Fumagalli et al, 2016; Grumati et al, 2017; Smith et al, 2018). However, which ER-resident cargo receptor contributes to starvation-induced ER-phagy was not elucidated in vivo. In this study, we have identified a truncated isoform of FAM134B (FAM134B-2) as a starvation-sensitive ER-resident cargo receptor. Starvation drastically induced FAM134B-2 but no other ER-resident cargo receptors, including FAM134B-1.

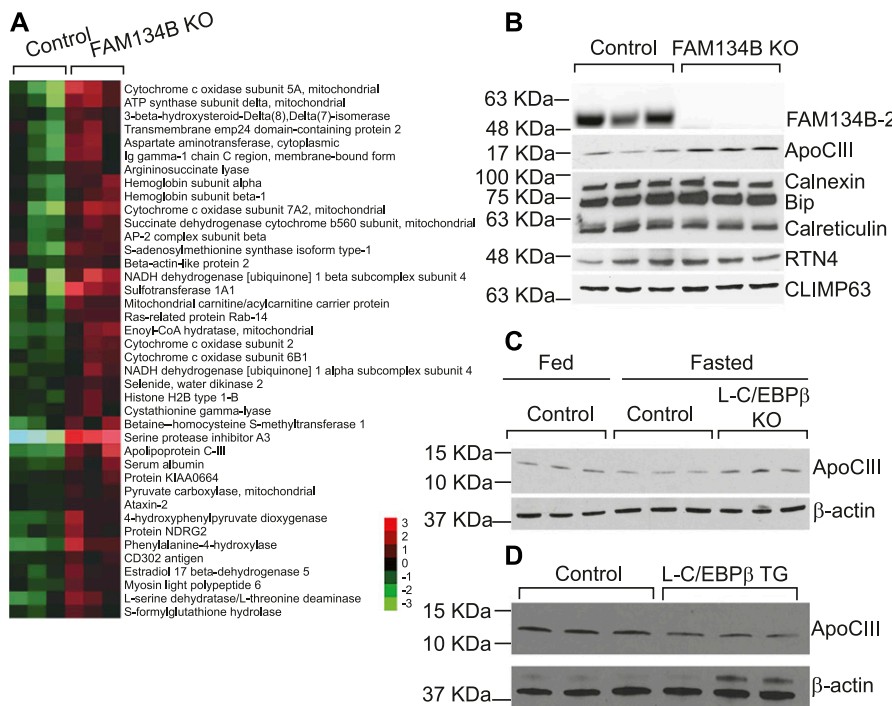

**Figure 6. FAM134B-2 induces starvation-induced ApoCIII degradation.**
**(A)** Heat map comparing protein levels in hepatic microsomes of FAM134B KO mice under starvation. Mice were fasted for 16 h. The details of the proteomics analysis were described in the Materials and Methods section. **(B)** Immunoblot analysis of microsomal ApoCIII, ER chaperones, RTN4, and CLIMP63 in the livers from fasted wild-type and FAM134B KO mice. **(C)** Immunoblot analysis of ApoCIII in the livers of L-C/EBPβ KO mice under fasted conditions. **(D)** Immunoblot analysis of ApoCIII in the livers of L-C/EBPβ TG mice under fed conditions.
Source data are available for this figure.

The induction of FAM134B-2 by starvation was observed not only in the liver but also in multiple peripheral organs, including the kidneys, spleen, and white adipose tissue. FAM134B-2 is localized in the ER and is capable of interacting with LC3B-II. In addition, the lysosomal degradation of FAM134B-2 was significantly increased under starvation. These results suggest that FAM134B-2 is a major ER-resident cargo receptor contributing to starvation-induced ER-phagy.

C/EBPβ (also called LAP) is a pleiotropic transcription factor that regulates cell differentiation, lipid metabolism, and gluconeogenesis (Rahman et al, 2007, 2016; Xu et al, 2013; Guo et al, 2015). We have identified C/EBPβ as a key regulator of starvation-induced FAM134B-2 expression and ER-related autophagy. Previously, more than 15 transcription factors were found to be regulators of fasting-induced hepatic gene expression (Goldstein & Hager, 2015). We tested 19 transcription factors known to control fasting-induced gene expression and autophagy gene expression in the regulation of *FAM134B-2* transcription. *FAM134B-2* promoter analysis revealed that C/EBPα and C/EBPβ strongly increased transcriptional activity. Consistently, ATG40, a putative yeast homologue of FAM134B, was induced by a starvation-mimic reagent, rapamycin, which alters C/EBPβ expression (Begay et al, 2015; Mochida et al, 2015). We also identified a specific element of the *FAM134B-2* promoter that drives C/EBP-induced promoter activation. However, because starvation induced hepatic C/EBPβ but not C/EBPα expression, starvation increased the recruitment of C/EBPβ but not C/EBPα onto the *FAM134B-2* promoter, hepatic C/EBPβ overexpression strongly induced FAM134B-2 expression under the fed state, and liver *C/EBPβ* deficiency significantly blocked starvation-induced FAM134B-2 expression, we think that C/EBPβ is more critical than C/EBPα in the regulation of starvation-induced hepatic FAM134B-2 expression.

Hepatic C/EBPβ expression directly induced autophagy-related genes and proteins such as ATG5 and LC3B-II. Furthermore, we were able to replicate starvation-induced FAM134B-2 expression in vitro using a human cell line, HeLa, by treating cells with a starvation media, EBSS, which is commonly used for inducing autophagy. Both in vivo and in vitro, starvation and C/EBPβ overexpression induced the recruitment of FAM134B-2 into autophagosomes and lysosomal FAM134B-2 degradation. Starvation induced lysosomal FAM134B-2 degradation, which was increased by bafilomycin A1 and leupeptin treatment in vitro and in vivo, respectively. In HeLa cells, C/EBPβ overexpression remarkably increased not only expression of FAM134B-2 but also the colocalization of FAM134B-2 and LC3. These data suggest that C/EBPβ regulates FAM134B-2–mediated ER-related autophagy. We need to determine which nutrients and hormones mediate the activation of C/EBPβ in the regulation of FAM134B-2–mediated ER-related autophagy. Previous reports showed that FXR, PPARα, and CREB coordinately regulate fasting-induced hepatic autophagy through the transcriptional regulation of autophagy genes including LC3 and ATG5 (Lee et al, 2014; Seok et al, 2014). However, PPARα and CREB overexpression did not affect *FAM134B-2* promoter activity. In addition, treatment of mice with a PPARα agonist (GW7647) and *FXR* deficiency did not affect hepatic *FAM134B-2* expression, whereas a known PPARα target, *FGF21*, was significantly increased by GW7647 (Fig S9A–C).

FAM134B-2 localizes in the SER and RER, whereas another ER-phagy receptor, SEC62, is specifically localized in the RER. These results suggest that whereas SEC62 specifically regulates hepatic RER-related autophagy, FAM134B2 contributes to both SER- and RER-related autophagy. In this study, we were unable to examine whether FAM134B-1 and FAM134B-2 localize in different regions of

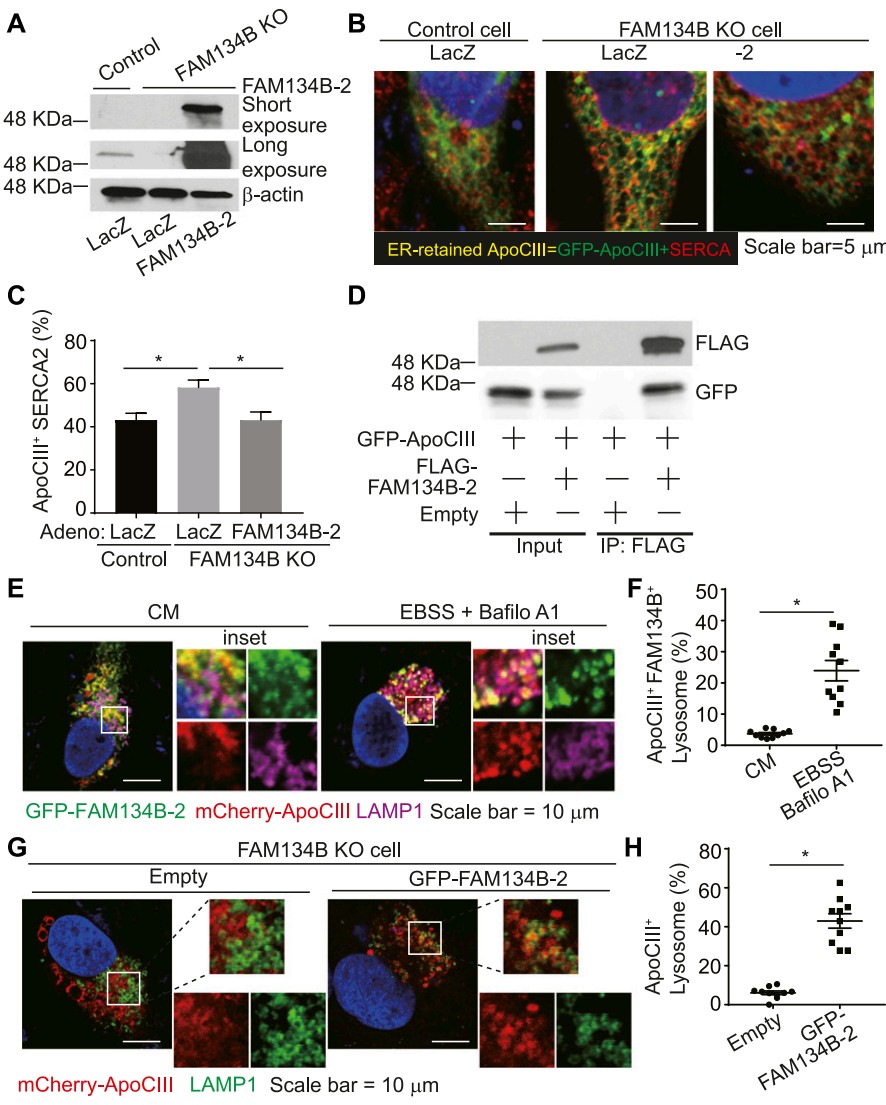

**Figure 7.  FAM134B-2 induces starvation-induced ApoCIII degradation.**
**(A–C)** Quantification of colocalization of EGFP-ApoCIII and SERCA2 positive ER. FAM134B KO HeLa cells were infected with adenovirus containing mouse FAM134B-2 and transfected with pEGFP-ApoCIII. **(A)** Immunoblot analysis of FAM134B-2 in HeLa FAM134B KO cells. **(B)** Representative images of HeLa FAM134B KO cells. **(C)** Quantification of GFP-ApoCIII positive areas in SERCA2-positive ER. **(D)** Co-immunoprecipitation of endogenous GFP-ApoCIII, FLAG-FAM134B-2. FLAG-FAM134B-2 was transiently overexpressed in HEK293T cells. 24 h after transfection, the cells were starved for 6 h in the presence of EBSS and bafilomycin A1. **(E, F)** Quantification of GFP-FAM134B-2 and mCherry-ApoCIII double positive LAMP1. The cells were transfected with GFP-FAM134B-2 and mCherry-ApoCIII. 24 h after transfection, the cells were starved for 6 h in the presence of EBSS and bafilomycin A1 or cultured with complete media. **(E)** Representative images of HeLa cells. **(F)** Quantification of GFP-FAM134B-2 and mCherry-ApoCIII double positive LAMP1. **(G, H)** Quantification of mCherry-ApoCIII–positive LAMP1. FAM134B KO HeLa cells were transfected with GFP-FAM134B-2 and mCherry-ApoCIII. 24 h after transfection, the cells were starved for 6 h in the presence of EBSS and bafilomycin A1. **(G)** Representative images of HeLa cells. **(H)** Quantification of mCherry-ApoCIII–positive LAMP1. One-way ANOVA with a Student–Newman post hoc test was used for statistical analysis. *$P < 0.05$.

the ER or examine the subcellular localization of other ER-resident cargo receptors such as full-length RTN3 and CCPG1 because of the lack of hepatic expression.

Although FAM134B-1 was originally identified as an ER-phagy receptor, recent studies suggest that FAM134B-1 regulates selective ER-phagies such as the degradation of protein procollagens, a native secretary protein family, which were identified as endogenous substrates for FAM134B-1 in fibroblasts (Forrester et al, 2018) and a Niemann-Pick C1 mutant (NPC1-I1061T) synthesized and modified in the ER (Schultz et al, 2018). We performed the functional analysis of FAM134B-2 to test whether FAM134B-2 regulates ER-phagy. FAM134B deficiency and FAM134B-2 reconstitution do not change the size of the ER and/or ER structure proteins in HeLa cells and mouse livers, suggesting that FAM134B-2 does not participate in bulk ER turnover. We next examined whether FAM134B-2 regulates selective ER protein degradation using a nontargeted proteomics approach in the hepatic microsome (crude ER) fraction of FAM134B KO mice. Interestingly, none of the ER structure proteins or ER chaperones were changed in

the livers of FAM134B KO mice. Most of the identified proteins are either secretary proteins (i.e., ApoCIII, SERPINA3, albumin, and hemoglobins) or mitochondrial proteins (i.e., cytochrome oxidases and ATP synthase). The ER is a critical site for the secretory protein system (Benham, 2012). ApoCIII is a component of triglyceride-rich lipoproteins such as VLDL (Khetarpal et al, 2017). In this study, we identified ApoCIII as one of the hepatic substrates for FAM134B-2–mediated selective ER-phagy. Co-immunoprecipitation showed that FAM134B-2 physically interacted with ApoCIII. Starvation reduced levels of ER-retained ApoCIII in an ATG7-dependent manner and increased the delivery of FAM134B-2–ApoCIII complexes to lysosomes. FAM134B deficiency failed to deliver ApoCIII to lysosomes, resulting in a significant accumulation of ApoCIII in the ER, whereas FAM134B-2 reconstitution to FAM134B KO cells normalized the delivery of ApoCIII to lysosomes. In addition, hepatic C/EBPβ deficiency blocked starvation-induced FAM134B-2, leading to the accumulation of ApoCIII, whereas C/EBPβ overexpression reduced levels of hepatic ApoCIII.

Previous studies exclusively examined the role of FAM134B-1 in the regulation of ER-phagy, colon cancer, vascular dementia, viral infection, and neuropathy (Kurth et al, 2009; Kong et al, 2011; Kasem et al, 2014; Khaminets et al, 2015; Islam et al, 2018). FAM134B-1 expression is limited in the brain, spleen, and testis. Although many of the peripheral tissues express both FAM134B-1 and FAM134B-2 isoforms at very low levels under normal conditions, starvation drastically induces FAM134B-2 but not FAM134B-1 in the peripheral tissues through the transcriptional activation of C/EBPβ. Further studies are required for examining why two isoforms of FAM134B in addition to three other ER-resident LC3B-II–binding proteins are present in the mammalian ER and the physiological role of the C/EBPβ-FAM134B-2-ER-lysosome protein degradation pathway. We believe that the present study sheds light on novel insights into starvation-induced selective ER-phagy.

## Materials and Methods

### Animals

To generate liver-specific C/EBPβ TG mice, we introduced the transgene at the Rosa26 locus by the recombinase-mediated cassette exchange method that we recently developed to circumvent the inherent problem of random insertion via traditional pronuclear injection (Masuda et al, 2016; Shiozaki et al, 2018). In brief, we transfected $5 \times 10^6$ R26FNF3-1F1 ES cells with 15 µg of pFLSLF3-Myc human C/EBPβ and 15 µg of pCAG-Flpe (plasmid 13787; Addgene). From the 96 clones tested, we found three clones with the expected genotype, which were also G418 sensitive, indicating successful exchange of the original neo cassette for the transgene cassette at the Rosa26 locus. Karyotypically normal ES clones were microinjected into C57BL/6 blastocysts to produce chimeric founders at the University of Colorado Bioengineering Core Facility. The generated mice were named Rosa26-C/EBPβ conditional TG (*Rosa26-C/EBPβ^loxtg/+*) mice. All mouse strains were backcrossed more than 10 times with C57Bl/6J mice. The genetic backgrounds were checked at the Bio-Resources Core Facility of the Barbara Davis Center at the University of Colorado-Denver. The *Rosa26-C/EBPβ^loxtg/+* mice were intercrossed with Albumin (Alb)-Cre mice to obtain *Alb-Cre*; *C/EBPβ^loxtg/+* mice and control *Alb-Cre* mice. To generate liver-specific *C/EBPβ* KO mice, *C/EBPβ*–floxed mice were purchased from Mutant Mouse Resource and Research Center (MMRRC, 034760-UNC) (Sterneck et al, 2006). The *C/EBPβ*–floxed mice were intercrossed with Alb-Cre mice to obtain *Alb-Cre(+)*; *C/EBPβ^flox/flox* mice and control *Alb-Cre(–)*; *C/EBPβ^flox/flox* mice. To generate *FAM134B* knockout mice, we flanked the last exon of *FAM134B* by two *lox*P sites (Fig S6D). In brief, a 2,487-bp DNA fragment encompassing the last exon of *FAM134B* was amplified by PCR using the BAC clone RP23-475N20 as a template. This DNA fragment was then cloned in between two *lox*P sites in the cloning vector pLFNFL at the *Pac*I site immediately upstream of an FRT-PGK-neo-polyA-FRT selection cassette to produce pL9FNFL. A left homologous arm encompassing 3,386 bp of DNA sequence upstream of exon 9 was PCR-amplified and cloned at the *Cla*I site, upstream of the proximal *lox*P site in pL9FNFL. Subsequently, a right homologous arm containing 2,703 bp of DNA sequence downstream of exon 6 was

PCR amplified and cloned at the *Not*I site, downstream of the distal *lox*P site. The resulting targeting construct was designated as pFam134b-LFNFL (13,535 bp). The targeting vector was linearized and introduced by electroporation into murine B6/129 hybrid EC7.1 embryonic stem cells. Karyotypically normal ES clones were microinjected into C57BL/6 blastocysts to produce chimeric founders at the Bioengineering Core facility in the University of Colorado-Denver. To generate *FAM134B KO* mice, the *FAM134B* lox mice were crossed with Rosa-Flp and EIIa-Cre mice and backcrossed 10 times with C57Bl6 mice (Fig S7A). 8-wk-old male C57BL6/J (WT) mice were purchased from the Jackson Laboratory. For the fasting and re-feeding study, WT mice were randomly separated to fed and fasted groups. After 16 h of fasting, mice were intraperitoneally injected with either vehicle or leupeptin (15 mg/kg body weight) (Gold Biotechnology) (Ezaki et al, 2014). Some of the fasted mice were re-fed a chow diet for an additional 6 or 16 h after fasting. Animal experiments were approved by the Institutional Animal Care and Research Advisory Committee of the University of Colorado at Denver.

### Cell cultures

HEK293T cells (ATCC), C2C12 cells (ATCC), HepG2 cells (ATCC), and HeLa cells (ATCC) were grown and maintained at 37°C in an atmosphere of 5% $CO_2$ in DMEM containing 4,500 mg/l glucose, 100 U/ml penicillin, and 100 µg/ml streptomycin and supplemented with 10% FBS. Mouse primary hepatocytes were isolated as described previously (Nakatani et al, 2002) and cultured in DMEM containing 1,000 mg/l glucose, 100 U/ml penicillin, and 100 µg/ml streptomycin and supplemented with 10% FBS. For starvation experiments, the cells were starved with EBSS and treated with 1 µM bafilomycin A1 or vehicle for 6 or 8 h.

### 5′RACE

5′RACE was performed using the SMARTer RACE 5′/3′ kit (Takara) according to the manufacturer's instructions. cDNA was prepared with the SMARTer RACE 5′/3′ Kit using 1 µg of RNA from mouse liver or brain. cDNA was amplified with mouse *FAM134B*-specific reverse primers, shown in Table S1. 5′RACE products were cloned into pGEM-T Easy vector (Promega) and sequenced.

### RNA analysis

qRT-PCR assays were performed using an Applied Biosystems StepOne Plus qPCR instrument. Quantitative expression values were calculated by the absolute standard curve method using plasmid template (Open Biosystems) containing each target gene cDNA. Primer sequences are listed in Table S1.

### Protter

Amino acid sequences of FAM134B-2 (NP_001264244.1) were obtained from National Center for Biotechnology Information. Transmembrane topology of FAM134B-2 was predicted and visualized using Protter ver.1.0 (Omasits et al, 2014).

## Transfections and luciferase assays

HEK293T cells were plated in 24-well plates and grown overnight. The cells were transfected with 300 ng firefly luciferase reporter plasmids containing 1.5-kb mouse *FAM134B-2* promoter (pGL3; Promega), 300 ng expression plasmid, and 50 ng of pCMV-LacZ vector using Turbofect (Thermo Fisher Scientific). After 24 h of transfection, the cells were harvested with 100 $\mu$l passive lysis buffer, and luciferase activities were measured using a luciferase assay system (Promega). Firefly luciferase activity was divided by $\beta$-galactosidase activity to obtain a normalized value, the relative luciferase unit.

## ChIP analysis

The samples were prepared using SimpleChIP Enzymatic Chromatin IP Kit (Magnetic Beads) from Cell Signaling Technology. ChIP was performed using rabbit IgG control C/EBP$\alpha$ or C/EBP$\beta$ antibody. Eluted DNA was analyzed by qRT-PCR. The results are expressed as the percentage of antibody binding versus the amount of PCR product obtained using a standardized aliquot of input chromatin (% of input). The primers for the C/EBP composite region of the mouse *FAM134B-2* promoter are shown in Table S1.

## ER fractionation

ER fractionation was performed using an ER isolation kit (ER0100; Sigma-Aldrich) according to the manufacturer's instructions. Microsomes isolated from 16-h fasted mouse livers were subjected to 15–30% Opti-Prep gradient and centrifuged at 150,000 *g* for 3 h at 4°C. After centrifugation, 15 fractions were collected from the top to the bottom of the gradient and subjected to immunoblot analysis.

## Immunoblot analysis

Cell and tissue lysates were prepared using RIPA buffer (150 mM NaCl, 1% Nonidet P-40, 0.5% sodium deoxycholate, 0.1% SDS, and 50 mM Tris, pH 8.0). The samples were separated by SDS–PAGE, transferred to a nitrocellulose membrane, and immunoblotted with the primary antibody. The samples were visualized using horseradish peroxidase coupled to appropriate secondary antibodies with enhancement by an ECL detection kit (Thermo Fisher Scientific). Primary antibodies are listed in Table S2. The FAM134B rabbit polyclonal antibody was produced against a peptide ((C) LPTELKRKKQQLDSAHR, amino acids 352–368; NP_001030023.1). The specificity of the FAM134B antibody was tested with immunoblot analysis using protein lysates from primary hepatocytes and C2C12 myoblasts treated with *FAM134B* siRNA (s82925; Thermo Fisher Scientific) as shown in Fig S10. The antibody recognized both mouse FAM134B-1 and FAM134B-2.

## Immunofluorescence

HeLa cells were grown on chamber slides, and after each experiment, the cells were fixed with 3% paraformaldehyde for 15 min. The cells were permeabilized with 0.05% digitonin in PBS for 10 min and blocked with 0.1% gelatin in PBS for 30 min. Subsequently, the cells were incubated with primary antibody solution for 1 h, washed in the permeabilizing buffer, and then incubated with secondary antibody

solution for 1 h. Coverslips were washed three times and mounted on microscope slides using Vectashield mounting medium with DAPI (Vector Laboratory). Fluorescence signals were captured using an Olympus FV1000 FCS/RICS (Olympus). To quantify the colocalization of FAM134B and LC3B, PCCs from at least 20 cells were calculated using NIH ImageJ. Primary antibodies are listed in Table S2.

## ER size quantitation

HeLa cells stably expressing mCherry-ER, mCherry-SEC61B, and HA-RTN4 were infected with adenovirus containing mouse FAM134B-2 and then treated with *FAM134B* siRNA (s29014; Thermo Fisher Scientific). Adenovirus containing mouse FAM134B-2 and lentivirus containing ER-mCherry (#55041; Addgene) were generated using the Gateway cloning system (Invitrogen). ER area measurements (20 cells) were conducted using NIH ImageJ as follows. Background threshold was manually defined and set for all images. Borders of each cell were drawn and the ER area was then calculated and presented as an ER-fraction of the total cell area.

## ER ApoCIII accumulation

HeLa cells lacking *FAM134B* or *ATG7* were generated by the CRISPR-Cas9 system as previously described (Shiozaki et al, 2018). The sgRNA target sequences are shown in Table S1. Wild-type and *FAM134B* KO HeLa cells expressing GFP-ApoCIII were infected with adenovirus containing mouse FAM134B-2 or LacZ (control). Lentivirus containing GFP-ApoCIII was generated using the Gateway cloning system (Invitrogen). The cells were immunolabeled with anti-SERCA2 antibody as described in the immunofluorescence section. Fluorescence signals were captured using an Olympus FV1000 FCS/RICS (Olympus). GFP-ApoCIII–positive SERCA2 areas from 20 cells were calculated using NIH ImageJ. Background threshold was manually defined and set for all images.

## Co-immunoprecipitation

FLAG-mouse FAM134B-2, FAM134B2-ΔLIR, and GFP-ApoCIII overexpressed HEK293T cells were starved with EBSS for 6 h in the presence of bafilomycin A1 (1 $\mu$M). Cell lysates were prepared using co-immunoprecipitation buffer (150 mM NaCl, 1% Nonidet P-40, and 50 mM Tris, pH 8.0). 2 mg of lysate and 40 $\mu$l of anti-DYKDDDDK G1 Affinity Resin (GenScript) were incubated overnight at 4°C. Resins were washed three times with co-immunoprecipitation buffer. Immunoprecipitated proteins were analyzed using immunoblot. The plasmid containing FAM134B2-ΔLIR was generated by replacing the LIR sequence (DDFELL) with alanines using a Q5 Site-Directed Mutagenesis Kit (New England Biolabs).

## Nontargeted proteomics

Hepatic microsome fractions (N = 3) were isolated using sequential ultracentrifugation as previously described (Man et al, 2006; Masuda et al, 2015). Proteomics was performed at the University of Colorado Mass Spectrometry Core Facility. The samples were digested using a modified filter-aided sample preparation method. Briefly, the samples were denatured, reduced, and alkylated using 4% SDS in 100 mM Tris–HCl buffer at pH 8.5, tris(2-carboxyethyl)phosphine (TCEP), and

iodoacetamide, respectively. SDS was removed using an Amicon Ultra 30-kD molecular weight cutoff spin filter by sequentially washing with 100 mM Tris–HCl, pH 8.5, containing 8 M urea followed by 2 M urea. The protein was then digested at room temperature for 4 h with a ratio of 1:50 Lys-C (Wako chemicals) to substrate and then overnight at 37°C with a ratio of 1:40 trypsin to substrate. The resulting peptide solutions were recovered by centrifugation at 14,000 $g$ for 10 min and washed twice with 80 $\mu$L of 0.5 M NaCl solution. The samples were desalted using Pierce C18 spin columns, according to the manufacturer's instructions. Desalted peptide samples were then evaporated to dryness in a Speed Vac concentrator at 45°C. The dried samples were resuspended in 3% ACN, 0.1% formic acid in water for LC–MS/MS analysis. Pooled samples were loaded into a 2 cm PepMap 100, nanoviper trapping column and chromatographically resolved online using a 0.075 × 250 mm, 2.0 $\mu$m Acclaim PepMap RSLC reverse-phase nanocolumn (Thermo Fisher Scientific) using a 1,290 Infinity II LC system equipped with a nanoadapter (Agilent). Mobile phases consisted of water + 0.1% formic acid (A) and 90% aq. acetonitrile + 0.1% formic acid (B). The samples were loaded into the trapping column at 3.2 $\mu$l/min for 2.5 min at an initial condition before being chromatographically separated at an effective flow rate of 330 nl/min using a gradient of 3–8% B over 4 min, 8–27% B over 101 min, and 27–40% B over 15 min for a 120-min total gradient at 42°C. The gradient method was followed by a column wash at 70% B for 5 min. Data were collected on a 6550 Q-TOF equipped with a nano-source (Agilent) operated using intensity-dependent CID MS/MS to generate peptide identifications. The capillary voltage, drying gas flow, and drying gas temperature were set to 1,300 V, 11.0 liters/min, and 175°C, respectively. MS/MS data were collected in positive ion polarity over mass ranges 290–1,700 m/z at a scan rate of 10 spectra/s for MS scans and mass ranges 50–1,700 m/z at a scan rate of 3 spectra/s for MS/MS scans. All charge states, except singly charged species, were allowed during MS/MS acquisition, and charge states 2 and 3 were given preference. SpectrumMill software (Agilent) was used to extract, search, and summarize peptide identity results. Spectra were searched against the SwissProt Mus Musculus database, allowing up to two missed tryptic cleavages with fixed carbamidomethyl (C) and variable deamidated (N), oxidation (M), and pyroglutamic acid (Q) modifications. The monoisotopic peptide mass tolerance allowed was ± 20.0 ppm and the MS/MS tolerance was ± 50 ppm. A minimum peptide score of 8 and scored peak intensity of 50% were used for generation of the AMRT library. The entire proteomics data are shown in Table S3.

## Statistics

Data were collected from more than two independent experiments and are reported as the mean ± SEM. Statistical analysis was performed using a two-tailed $t$ test for two-group comparison and a one-way ANOVA with a Student–Newman post hoc test or two-way ANOVA for multigroup comparison. Significance was accepted at $P < 0.05$.

## Supplementary Information

## Acknowledgements

The authors' work was supported by grants from National Institutes of Health R01DK096030, R01HL117062, R01HL133545, and R01HL132318 to M. Miyazaki.

### Author Contributions

S Kohno: conceptualization, data curation, software, formal analysis, validation, investigation, visualization, methodology, and writing—original draft.
Y Shiozaki: conceptualization, resources, and software.
AL Keenan: data curation, formal analysis, investigation, methodology, and writing—original draft, review, and editing.
S Miyazaki-Anzai: resources, data curation, formal analysis, investigation, visualization, and methodology.
M Miyazaki: conceptualization, resources, data curation, software, formal analysis, supervision, funding acquisition, validation, investigation, visualization, methodology, project administration, and writing—original draft, review, and editing.

### Conflict of Interest Statement

The authors declare that they have no conflict of interest.

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
