## [Reviewer comments · Life Science Alliance]

Life Science Alliance

A truncated isoform of FAM134B (FAM134B-2) regulates starvation-induced hepatic selective ER-phagy

Shohei Kohno, Yuji Shiozaki, Audrey Keenan, Shinobu Miyazaki-Anzai, and Makoto Miyazaki
DOI: <https://doi.org/10.26508/lsa.201900340>

Corresponding author(s): Makoto Miyazaki, University of Colorado Denver

Review Timeline:	Submission Date:	2019-02-11
	Editorial Decision:	2019-02-28
	Revision Received:	2019-04-15
	Editorial Decision:	2019-04-30
	Revision Received:	2019-05-07
	Accepted:	2019-05-07

Scientific Editor: Andrea Leibfried

Transaction Report:

February 28, 2019

Re: Life Science Alliance manuscript #LSA-2019-00340

Makoto Miyazaki
University of Colorado Denver

Dear Dr. Miyazaki,

Thank you for submitting your manuscript entitled "An N-terminal truncated isoform of FAM134B (FAM134B-2) regulates starvation-induced selective ER-phagy" to Life Science Alliance. The manuscript was assessed by expert reviewers, whose comments are appended to this letter.

As you will see, the reviewers appreciate your data and provide constructive input on how to further strengthen it. We would thus like to invite you to provide a revised version of your work, addressing the concerns of the three reviewers. Importantly, the selective ER-phagy of AopCIII needs further validation (reviewer #1 and #3) and the specific localization of FAM134b-2 needs to get re-assessed (all three reviewers) as well as its size (reviewer #2) and the effect of its deficiency on the size of the ER (reviewer #3).

Thank you for this interesting contribution to Life Science Alliance. We are looking forward to receiving your revised manuscript.

Sincerely,

Andrea Leibfried, PhD
Executive Editor

Life Science Alliance
Meyerhofstr. 1
69117 Heidelberg, Germany
t +49 6221 8891 502
e a.leibfried@life-science-alliance.org
www.life-science-alliance.org

B. MANUSCRIPT ORGANIZATION AND FORMATTING:

Reviewer #1 (Comments to the Authors (Required)):

The manuscript entitled "An N-terminal truncated isoform of FAM134B (FAM134B-2) regulates starvation induced selective ER-Phagy" proposed the identification of a novel N-terminal truncated isoform of FAM134B, as a starvation-induced selective cargo receptor in vivo. Specifically they elucidate a transcriptional regulation of FAM134B-2 via the transcription factor C/EBP β in the liver. The manuscript is overall well written with a clear take-home message, the experiments are well presented with a good flow and rationale. The experiments in vivo are elegant and clear. The

concept of ER-phagy as a degradative pathway that mediates APOCIII is weak, and the molecular mechanism is missing. However some minor and a few major comments are detailed below:

Major comments

- In figure 7, the authors identified cargoes for a selective FAM134B-2 mediated ER-phagy of 40 secretory proteins. It is not clear to me why they have focused on ApoCIII.
- The images in figure 7E are unclear, and not representative of the quantification graph in figure 7F. Please change them accordingly.
- The authors proposed that the expression of FAM134B-2 is important to mediate selective ER-Phagy of ApoCIII, however to further validate these data the authors should perform colocalization IF experiments of ApoCIII-FAM134b-2- LC3 and/or lysosome in starved wt and Fam134bKO cells in presence of lysosome inhibitor (such as leupeptin). Furthermore they could take advantage also from the use of FAM134b-2 delta LIR construct to show no delivery of ApoCIII to the autophagosome/lysosome.
- Does FAM134b-2 immunoprecipitates ApoCIII?
- In figure 7B-C the authors showed that FAM134bKO livers accumulated ApoCIII, as well as L-C/EBPb KO livers. Does the L-C/EBPb tg mice, show a increased degradation of ApoCIII.
- The assumption that FAM134b-2 is localized in the SER is not really convincing (FIG5a). Also, SER is deputed to synthesize lipids, however in figure 7A, they showed by proteomics that secretory protein (i.e. serum albumin) accumulated in microsomes of Fam134bKO. Since RER subdomain controls secretory pathway, could the authors explain on the finding that Fam134b-2 localizes in the SER but its cargoes are secretory proteins?

Minor comments

- in qRT-PCR experiments please specify "fold change relative to.." in the graphs.
- In Fig 6E and H, if possible provide higher resolution and more zoomed images with single channel insets.
- In figure S2B, please provide a schematic representation of the expected PCR fragment size and primers designed on the sequence.
- Please provide reference for the generation of Myc-CEBPb-Rosa26 tg mice.
- RTN3 and CCPG1, please be consistent between the results section and the discussion (are they expressed at low level or do the antibodies not work properly?)
- Regarding human in vitro cell line, the HepG2 do not recapitulate the in vivo finding. Have the authors tried alternative human hepatocellular carcinoma cell line (such as HuH7)?

Reviewer #2 (Comments to the Authors (Required)):

In this study, Kohno et al have characterized a novel isoform of FAM134B in the liver and other organs, FAM134B-2 which regulates ER-phagy. This isoform is truncated on the N-terminus and lacks its reticulon like function but retains its LC3-binding region and activity. The authors have identified the promotor region and transcription factor (C/EBP β) required for FAM134B-2

transcription and have shown that ApoCIII is a target for degradation through FAM134B-2. The study represents an advance in the field, since it expands on our knowledge of ER-phagy and how FAM134B could play a role in that. Specifically the fact that the authors characterize this response in an in vivo system is relevant.

Major points:

1. The title should be amended to reflect the organ specific expression of FAM134B-2.

2. Fig 2A: More explanation/experimental evidence is needed to investigate the FAM134B-2 protein. There are a lot of bands on the gel, and the explanation of the authors of post translational modifications is not enough.

A few crucial points that need clarifying are: -Why is FAM134B in the brain smaller than the expressed flag-FAM134B? If the authors speculate that the FLAG tag could induce such a shift, even though the predicted increase for the flag epitope is only about 1 KDa, the authors should try an alternative tag. -There are (faint) bands visible in the fed condition in mouse liver. What are these? They are clearly higher than the FAM134B-2 signal in fasted condition, but also higher than in refed conditions. How can this be?

Depending on the solution the authors put forward, this can be done in a month, a few at most.

3. The authors have too much "data not shown" in the manuscript (6 times). In almost all of these cases the data would actually have been interesting to see. A case in point is the comparison between human and mouse FAM134B-2, this is informative enough to warrant inclusion. Another example is the observation that FAM134B-2 mRNA is increased after starvation in HEK293T cells, but not in the other cell lines. Please show all this data.

4. Fig. 3: The authors state that they are probing the transcription factor that specifically binds the FAM134B-2 promotor, but this will be the same promotor as FAM134B-1 since it is transcribed from the same DNA as an isoform. This should be stated in the text. As far as I understand it, C/EBP β thus increases transcription of FAM134B-1 as well. The authors should rule this out or alter the text to state this. If the authors decide on the second option, that would still leave the question as to why in the liver this specific isoform is expressed unanswered. Ideally, if the authors already have an idea why perhaps some key experiments could greatly aid the mechanistic insights into FAM134B-2, otherwise this could just be expanded on in the discussion.

5. Fig 4: The conclusion that FAM134B-2 is localized only to the SER is too strong. When looking at the blot, it is clear that there is a strong signal for SEC62, which the authors state is their marker for RER in lane number 9. Lane number 9 is also the lane which has a strong band (strongest band?) for FAM134B-2. This is in conjunction with Fig S4. The authors should also perform a colocalization analysis between FAM134B-2 and SEC62 to see if there is much difference. Otherwise the authors should alter the text.

6. Fig 6: The size of the FAM134B protein is crucial to judge whether it is the 1 or the 2 isoform, all protein markers should be clearly and accurately displayed on all blots. The authors should also show a full immunoblot, as in figure S2C of all cell lines used so the readers can judge all the bands in all the conditions tested. In Khaminets et al (cited multiple times by the authors) FAM134B is expressed in U2OS cells, so this should be visible on this immunoblot. The authors should also explain more clearly and with more detail why each cell line was chosen. Otherwise, it seems as if the authors used standard cell lines that were present in the lab to see which one fit their model.

Minor points:

1. First 5 lines of results are actually introduction. They should be integrated there.
2. Figure 1 seems a little light showing only the current data. I would suggest merging with figure 2.
3. Fig. 2B vs Fig S2B: Why is the band of FAM134B-2 lower than FAM134B-1 in Fig. 2B but higher than FAM134B-1 in Fig. S2B?
4. C/EBP β is introduced suddenly in Fig. S2C without any prior explanation (the explanation only comes afterwards). This should be addressed to make it clearer why the authors are looking at it at this point.
5. Fig S6B: The intensity of the mCherry-ER seems higher in the FAM13B-2 over expressing cells. Why is that? Does the ER have a higher capacity for protein translation?
6. Fig 6G: LC3 spots should be counted per cell with a program like Imaris. Intensity could work only as a secondary marker.

Reviewer #3 (Comments to the Authors (Required)):

Summary

In this manuscript, Kohno et al. set out to define which selective autophagy receptor mediates starvation-induced ERphagy in the liver. As a starting point the authors monitored mRNA expression of the four known ERphagy receptors in livers from fed, fasted and re-fed mice and found that FAM134B is exclusively induced. Subsequent FAM134B immunoblot and RACE analyses unveiled that an N-terminally truncated isoform (FAM134B-2) is expressed in livers from fasted mice instead of the full-length protein (FAM134B-1). FAM134B-2 shared with FAM134B-1 the C-terminal part that contains the LIR motif and retained the ability to bind LC3B. To dissect the signaling axis which lead to the upregulation of FAM134B-2, the authors screen a focused set of transcription factors and identified C/EBP β as transcriptional inducer of FAM134B-2. C/EBP β was found to bind the FAM134B-2 promoter and to be upregulated in livers of fasted mice. Consistently, liver-specific transgenic C/EBP β mice showed increased levels of FAM134B-2 under fed conditions, while expression of FAM134B-2 was almost ablated in fasted mice with conditional C/EBP β deletion in the liver. Using subcellular fractionation of mice liver, the authors showed that FAM134B-2 was present in distinct ER fractions and sensitive to lysosomal protease inhibitors. The authors recapitulated their in vivo findings in HeLa cells including FAM134B-2 and C/EBP β induction upon starvation and localization of FAM134B-2 with ER markers, LC3B and LAMP1. Next, the authors demonstrated that the abundance of the secreted protein ApoCIII is dependent by FAM134B-2 and C/EBP β . Together, this is a fascinating study which provides compelling in vivo evidence for a liver specific FAM134B variant that confers hepatic starvation-induced ERphagy. However, several concerns remain.

Major point

- 1) Almost all immunoblots are missing molecular weight markers. The authors should provide this information in each case.
- 2) The same holds true for scale bars in microscopy images.
- 3) The authors' conclusion that FAM134B-2 is "is only detected in SER fractions" is not supported by the immunoblot analysis (Figure 5). In fact, the distribution of FAM134C and FAM134B-2 overlaps with that of SER and RER markers. Moreover, the confocal microscopy in Figure S4 is lacking a comparison with an RER marker. The authors should perform additional experiments (e.g.

quantitative colocalization studies with RER and SER) to substantiate their claim.

4) The authors should provide biochemical evidence that FAM134B-2 and ApoCIII are delivered to lysosomes in an ATG5/ATG7-dependent manner. This can be done by protease protection assays, lysosome IP or lysosome enrichment protocols. Moreover, the authors should examine whether the abundance of ApoCIII is sensitive to ATG5/ATG7 deletion/depletion and BafA1 treatment.

5) The finding that FAM134B deficiency does not alter the size of the ER is in contrast to previous finding (e.g. Khaminets et al 2015). For their analysis, the authors use a CALR-based cherry-ER reporter. Given the ER architecture of sheets and tubules, the authors should use more appropriate reporter to assess morphological changes of the ER (e.g. CLIMP-63 and SEC61B as marker proteins of sheet-like cisternal ER and RTN4 as marker proteins for tubular ER). In this context, the authors should examine the re-expression of FAM134B-2 in FAM134B deficient cells

Reviewer #1 (Comments to the Authors (Required)):

The manuscript entitled "An N-terminal truncated isoform of FAM134B (FAM134B-2) regulates starvation induced selective ER-Phagy" proposed the identification of a novel N-terminal truncated isoform of FAM134B, as a starvation-induced selective cargo receptor *in vivo*. Specifically they elucidate a transcriptional regulation of FAM134B-2 via the transcription factor C/EBP β in the liver. The manuscript is overall well written with a clear take-home message, the experiments are well presented with a good flow and rationale. The experiments *in vivo* are elegant and clear. The concept of ER-phagy as a degradative pathway that mediates APOCIII is weak, and the molecular mechanism is missing. However some minor and a few major comments are detailed below:

Major comments

- In figure 7, the authors identified cargoes for a selective FAM134B-2 mediated ER-phagy of 40 secretory proteins. It is not clear to me why they have focused on ApoCIII.

We first chose Serpin A3 as a FAM134B target since it was most significantly increased in the livers of FAM134B KO mice. However, we were not able to find a good antibody against mouse Serpin A3. We therefore chose ApoCIII, which was second-most consistently regulated by FAM134B2 deficiency. More importantly, we were able to find an antibody that specifically detects mouse ApoCIII.

- The images in figure 7E are unclear, and not representative of the quantification graph in figure 7F. Please change them accordingly.

We replaced the image.

- The authors proposed that the expression of FAM134B-2 is important to mediate selective ER-Phagy of ApoCIII, however to further validate these data the authors should perform colocalization IF experiments of ApoCIII-FAM134b-2- LC3 and/or lysosome in starved wt and Fam134bKO cells in presence of lysosome inhibitor (such as leupeptin). Furthermore they could take advantage also from the use of FAM134b-2 delta LIR construct to show no delivery of ApoCIII to the autophagosome/lysosome.

We included the additional data for the co-localization of ApoCIII-FAM134B-2-LAMP1 in the presence of starvation and Baf. A1 in Fig. 7E-H.

- Does FAM134b-2 immunoprecipitates ApoCIII?

Yes: we included the additional data in Fig. 7D.

- In figure 7B-C the authors showed that FAM134bKO livers accumulated ApoCIII, as well as L-C/EBPb KO livers. Does the L-C/EBPb tg mice, show a increased degradation of ApoCIII.

Yes: we included the additional data in Fig. 6D

- The assumption that FAM134b-2 is localized in the SER is not really convincing (FIG5a). Also, SER is deputed to synthesize lipids, however in figure 7A, they showed by proteomics that secretory protein (i.e. serum albumin) accumulated in microsomes of Fam134bKO. Since RER subdomain controls secretory pathway, could the authors explain on the finding that Fam134b-2 localizes in the SER but its cargoes are secretory proteins?

As other reviewers have also pointed this out, we agree that FAM134B-2 may not be preferentially localized in the SER. In addition, FAM134B-2 is co-localized with a RER-specific marker, SERC62. The description has been changed to "FAM134B-2 is localized in both the SER and RER".

Minor comments

- in qRT-PCR experiments please specify "fold change relative to.." in the graphs.

Added (such as vs. Fed).

- In Fig 6E and H, if possible provide higher resolution and more zoomed images with single channel insets.

We understand the resolution issues of the images. However, our current confocal microscopy system is unable to provide better images.

- In figure S2B, please provide a schematic representation of the expected PCR fragment size and primers designed on the sequence.

Added (Fig. S2C).

- Please provide reference for the generation of Myc-CEBPb-Rosa26 tg mice.

Added

- RTN3 and CCPG1, please be consistent between the results section and the discussion (are they expressed at low level or do the antibodies not work properly?)

Thank you for pointing this out. We currently have antibodies that detect mouse RNT3 and CCPG1. RTN3 and CCPG1 were undetectable due to their low expressions in the liver.

- Regarding human in vitro cell line, the HepG2 do not recapitulate the in vivo finding. Have the authors tried alternative human hepatocellular carcinoma cell line (such as HuH7)?

We have not tested HuH7. It looks like HuH7 is not commercially available in the US (ATCC). We would like to test it in a future study.

Reviewer #2 (Comments to the Authors (Required)):

In this study, Kohno et al have characterized a novel isoform of FAM134B in the liver and other organs, FAM134B-2 which regulates ER-phagy. This isoform is truncated on the N-terminus and lacks its reticulon like function but retains its LC3-binding region and activity. The authors have identified the promotor region and transcription factor (C/EBP β) required for FAM134B-2 transcription and have shown that ApoCIII is a target for degradation through FAM134B-2. The study represents an advance in the field, since it expands on our knowledge of ER-phagy and how FAM134B could play a role in that. Specifically the fact that the authors characterize this response in an in vivo system is relevant.

Major points:

1. The title should be amended to reflect the organ specific expression of FAM134B-2.

We included “hepatic” in the title, since we focused on the liver in this study.

2. Fig 2A: More explanation/experimental evidence is needed to investigate the FAM134B-2 protein. There are a lot of bands on the gel, and the explanation of the authors of post translational modifications is not enough.

A few crucial points that need clarifying are: -Why is FAM134B in the brain smaller than the expressed flag-FAM134B? If the authors speculate that the FLAG tag could induce such a shift, even though the predicted increase for the flag epitope is only about 1 KDa, the authors should try an alternative tag. -There are (faint) bands visible in the fed condition in mouse liver. What are these? They are clearly higher than the FAM134B-2 signal in fasted condition, but also higher than in refeed conditions. How can this be?

Depending on the solution the authors put forward, this can be done in a month, a few at most.

It is due to the fact that human FAM134B-1 is slightly bigger than mouse FAM134B-1, in addition to the FLAG tag. To circumvent this confusion, we used mouse FAM134B-1 and FAM134B-2 (without a tag) recombinant proteins as the markers and ran the WB again, as shown in the new Fig. 1B and 1G. The faint bands are non-specific signals. As you can see in the new blot, there was no significant change among the groups. In addition, since we did not detect the faint band in other blots of liver or mouse hepatocytes in Fig 6B, Fig S8E and Fig S10, this non-specific band appears depending on the lot of the FAM134B antibody.

3. The authors have too much "data not shown" in the manuscript (6 times). In almost all of these cases the data would actually have been interesting to see. A case in point is the comparison between human and mouse FAM134B-2, this is informative enough to warrant inclusion. Another example is the observation that FAM134B-2 mRNA is increased after starvation in HEK293T cells, but not in the other cell lines. Please show all this data.

All of "data not shown" were included as figures.

4. Fig. 3: The authors state that they are probing the transcription factor that specifically binds the FAM134B-2 promotor, but this will be the same promotor as FAM134B-1 since it is transcribed from the same DNA as an isoform. This should be stated in the text. As far as I understand it, C/EBP β thus increases transcription of FAM134B-1 as well. The authors should rule this out or alter the text to state this. If the authors decide on the second option, that would still leave the question as to why in the liver this specific isoform is expressed unanswered. Ideally, if the authors already have an idea why perhaps some key experiments could greatly aid the mechanistic insights into FAM134B-2, otherwise this could just be expanded on in the discussion.

We do not think that FAM134B-1 and FAM134B-2 share the promoter, since 1) the first exon of FAM134B-1 and FAM134-2 are different and 2) their transcription initiation sites are >10 kb apart.

5. Fig 4: The conclusion that FAM134B-2 is localized only to the SER is too strong. When looking at the blot, it is clear that there is a strong signal for SEC62, which the authors state is their marker for RER in lane number 9. Lane number 9 is also the lane which has a strong band (strongest band?) for FAM134B-2. This is in conjunction with Fig S4. The authors should also perform a colocalization analysis between FAM134B-2 and SEC62 to see if there is much difference. Otherwise the authors should alter the text.

As you suggested, we performed the co-localization analysis between FAM134B-2 and SEC62 (Fig. S4C). Since FAM134B-2 and SEC62 are co-localized, we changed the descriptions. FAM134B-2 is expressed in both the SER and RER.

6. Fig 6: The size of the FAM134B protein is crucial to judge whether it is the 1 or the 2 isoform, all protein markers should be clearly and accurately displayed on all blots. The authors should also show a full immunoblot, as in figure S2C of all cell lines used so the readers can judge all the bands in all the conditions tested. In Khaminets et al (cited multiple times by the authors) FAM134B is expressed in U2OS cells, so this should be visible on this immunoblot. The authors should also explain more clearly and with more detail why each cell line was chosen. Otherwise, it seems as if the authors used standard cell lines that were present in the lab to see which one fit their model.

The expanded picture of the immunoblot was added into Fig. 5C as the reviewer suggested. We also included HEK293 cells overexpressing human FAM134B-1 and FAM134B-2 as markers.

U2OS cells express FAM134B-1 at very low levels in contrast to the paper (Khaminets et al.) We tested FAM134B-2 expression in the cell lines that are most commonly used for autophagy research. As we described in the text, we looked for a cell line in which starvation induces FAM134B-2.

Minor points:

1. First 5 lines of results are actually introduction. They should be integrated there.

We have integrated the description into the Introduction.

2. Figure 1 seems a little light showing only the current data. I would suggest merging with figure 2.

Fig. 1 and Fig. 2 were merged

3. Fig. 2B vs Fig S2B: Why is the band of FAM134B-2 lower than FAM134B-1 in Fig. 2B but higher than FAM134B-1 in Fig. S2B?

Those analyses are different. Fig. 1C (previously Fig. 2B) shows data from the RACE analysis, whereas Fig. S2C (previously Fig. S2B) shows data from the PCR analysis.

4. C/EBP β is introduced suddenly in Fig. S2C without any prior explanation (the explanation only comes afterwards). This should be addressed to make it clearer why the authors are looking at it at this point.

The data for C/EBP β have been moved to Fig. S3A.

5. Fig S6B: The intensity of the mCherry-ER seems higher in the FAM13B-2 over expressing cells. Why is that? Does the ER have a higher capacity for protein translation?

There are no significant differences between mCherry-ER expression and FAM134B-2 overexpressing cells (currently Fig. S7B). We replaced it with a better image.

6. Fig 6G: LC3 spots should be counted per cell with a program like Imaris. Intensity could work only as a secondary marker.

C/EBP β significantly induced LCB3 puncta size as well, resulting in many of the puncta overlapping. Therefore counting the puncta number was impossible. We analyzed the LC3-

positive area instead of the intensity. We hope that the reviewer prefers the area more than the intensity.

Reviewer #3

Summary

In this manuscript, Kohno et al. set out to define which selective autophagy receptor mediates starvation-induced ERphagy in the liver. As a starting point the authors monitored mRNA expression of the four known ERphagy receptors in livers from fed, fasted and re-fed mice and found that FAM134B is exclusively induced. Subsequent FAM134B immunoblot and RACE analyses unveiled that an N-terminally truncated isoform (FAM134B-2) is expressed in livers from fasted mice instead of the full-length protein (FAM134B-1). FAM134B-2 shared with FAM134B-1 the C-terminal part that contains the LIR motif and retained the ability to bind LC3B. To dissect the signaling axis which lead to the upregulation of FAM134B-2, the authors screen a focused set of transcription factors and identified C/EBPbeta as transcriptional inducer of FAM134B-2. C/EBPbeta was found to bind the FAM134B-2 promotor and to be upregulated in livers of fasted mice. Consistently, liver-specific transgenic C/EBPbeta mice showed increased levels of FAM134B-2 under fed conditions, while expression of FAM134B-2 was almost ablated in fasted mice with conditional C/EBPbeta deletion in the liver. Using subcellular fractionation of mice liver, the authors showed that FAM134B-2 was present in distinct ER fractions and sensitive to lysosomal protease inhibitors. The authors recapitulated their in vivo findings in HeLa cells including FAM134B-2 and C/EBPbeta induction upon starvation and localization of FAM134B-2 with ER markers, LC3B and LAMP1. Next, the authors demonstrated that the abundance of the secreted protein ApoCIII is dependent by FAM134B-2 and C/EBPbeta. Together, this is a fascinating study which provides compelling in vivo evidence for a liver specific FAM134B variant that confers hepatic starvation-induced ERphagy. However, several concerns remain.

Major

point

1) Almost all immunoblots are missing molecular weight markers. The authors should provide this information in each case.

We have added MW markers to the figures with immunoblots.

2) The same holds true for scale bars in microscopy images.

We have added scales on the figures with microscopy images.

3) The authors' conclusion that FAM134B-2 is "is only detected in SER fractions" is not supported by the immunoblot analysis (Figure 5). In fact, the distribution of FAM134C and FAM134B-2 overlaps with that of SER and RER markers. Moreover, the confocal microscopy in Figure S4 is lacking a comparison with an RER marker. The authors should perform additional

experiments (e.g. quantitative colocalization studies with RER and SER) to substantiate their claim.

As other reviewers have also pointed this out, we agree that FAM134B-2 may not be specifically localized in the SER. In addition, FAM134B-2 is co-localized with an RER specific marker, SEC62 (Fig. S4C). The description has been changed to “FAM134B-2 is localized in both the SER and RER”.

4) The authors should provide biochemical evidence that FAM134B-2 and ApoCIII are delivered to lysosomes in an ATG5/ATG7-dependent manner. This can be done by protease protection assays, lysosome IP or lysosome enrichment protocols. Moreover, the authors should examine whether the abundance of ApoCIII is sensitive to ATG5/ATG7 deletion/depletion and BafA1 treatment.

ATG7 KO increased levels of FAM134B-2 and ApoCIII, and also induced accumulation of ApoCIII in the ER (Fig. S8H-S8J).

5) The finding that FAM134B deficiency does not alter the size of the ER is in contrast to previous finding (e.g. Khaminets et al 2015). For their analysis, the authors use a CALR-based cherry-ER reporter. Given the ER architecture of sheets and tubules, the authors should use more appropriate reporter to assess morphological changes of the ER (e.g. CLIMP-63 and SEC61B as marker proteins of sheet-like cisternal ER and RTN4 as marker proteins for tubular ER). In this context, the authors should examine the re-expression of FAM134B-2 in FAM134B deficient cells

We performed this additional study, but the size of both SEC61B-positive ER and RTN4-positive ER were not changed by FAM134B-2 modulation (Fig. S7E-S7H).

April 30, 2019

RE: Life Science Alliance Manuscript #LSA-2019-00340R

Prof. Makoto Miyazaki
University of Colorado Denver
12700 East 19th C281
Aurora, Colorado 80045

Dear Dr. Miyazaki,

Thank you for submitting your revised manuscript entitled "A truncated isoform of FAM134B (FAM134B-2) regulates starvation-induced hepatic selective ER-phagy". As you will see, the reviewers appreciate the introduced changes, though both reviewer #1 and #3 would have been even more positive if additional insight at subcellular level were provided into the autophagic process potentially at play for ApoCIII. We appreciate that you tested for ApoCIII levels in ATG7 KO cells and while we agree with reviewer #3 that further imaging analyses in autophagy-deficient cells would be helpful, adding such analysis is not mandatory for publication here. We would thus be happy to publish your paper in Life Science Alliance pending final revision:

- please include the proteomics results or deposit them in a suitable database (<http://www.life-science-alliance.org/manuscript-prep#datadepot>)
- please provide source data for the western blots in figure 3E and 6C
- please link your profile in our submission system to your ORCID iD, you should have received an email with instructions on how to do so

A. FINAL FILES:

B. MANUSCRIPT ORGANIZATION AND FORMATTING:

Sincerely,

Reviewer #1 (Comments to the Authors (Required)):

I am satisfied with the revision.

I still believe that fig.7 contains data that are not as good as the rest of the paper, but overall this work is a nice story and focuses on a very new topic.

Reviewer #2 (Comments to the Authors (Required)):

The authors have sufficiently addressed my comments and I would now recommend publication of this article titled "An N-terminal truncated isoform of FAM134B (FAM134B-2) regulates starvation induced hepatic selective ER-phagy." in Life Science Alliance.

Reviewer #3 (Comments to the Authors (Required)):

The authors adequately addressed all my concerns with the exception of showing that the delivery of ApoCIII to lysosomes is ATG5 and/or ATG7 dependent. While the authors indirectly show an involvement of ATG7 (e.g. altered abundance of ApoCIII), the experimental setting shown in Figure 7E/G could have been used to clarify this issue. Given that lysosomal delivery of cytosolic accessible content is the defining feature of macroautophagy, determining the molecular requirement of this process for a newly involved factor is of broad interest for the field since it will help classifying this new subtype in the seemingly never-ending array of selective autophagy processes. To increase the impact of this study, I would therefore highly recommend to perform this additional experiment.

May 7, 2019

RE: Life Science Alliance Manuscript #LSA-2019-00340RR

Prof. Makoto Miyazaki
University of Colorado Denver
12700 East 19th C281
Aurora, Colorado 80045

Dear Dr. Miyazaki,

Thank you for submitting your Research Article entitled "A truncated isoform of FAM134B (FAM134B-2) regulates starvation-induced hepatic selective ER-phagy". It is a pleasure to let you know that your manuscript is now accepted for publication in Life Science Alliance. Congratulations on this interesting work.

DISTRIBUTION OF MATERIALS:

Again, congratulations on a very nice paper. I hope you found the review process to be constructive and are pleased with how the manuscript was handled editorially. We look forward to future exciting submissions from your lab.

Sincerely,
